# Institutions and the resource curse: New insights from causal machine learning

**Roland Hodler[1,2,3,4], Michael Lechner[5,6,7], Paul A. Raschky [8]***

**1** Department of Economics and SIAW-HSG, University of St. Gallen, St. Gallen, Switzerland, **2** CEPR, London, United Kingdom, **3** CESifo, Munich, Germany, **4** OxCarre, University of Oxford, Oxford, England, **5** IAB, Nuremberg, Germany, **6** IZA, Bonn, Germany, **7** RWI, Essen, Germany, **8** Department of Economics and SoDa Labs Monash University, Caulfield East, VIC, Australia

* paul.raschky@monash.edu

## Abstract

There is a widely held belief that natural resource rents are a blessing if institutions are strong, but a curse if institutions are weak. We use data from 3,800 Sub-Saharan African districts and apply a causal forest estimator to reassess the relationship between institutions and the effects of resource rents. Consistent with this belief, we document that stronger institutions increase the positive effect of the presence of mining activities on economic development and dampen the negative effect of mining activities on conflict. In contrast, we find that the effects of higher world mineral prices on economic development and conflict in mining districts are non-linear and vary little in institutional quality.

## Introduction

The question of whether and when natural resources are a curse has attracted a lot of attention in the literature ever since the seminal work by [1]. In other prominent contributions, [2,3] focus on the role of institutions. They present a stylized model predicting that natural resource rents lead to more rent seeking and, consequently, lower levels of income if economic institutions are weak (or "grabber-friendly" as they call it), but lead to higher incomes if economic institutions are strong (or "producer-friendly"). In subsequent theoretical work [4,5], model the political process more explicitly and predict that natural resource rents lead to higher incomes if political institutions are strong, but to corruption, political violence, and lower incomes if political institutions are weak.

The general argument that natural resource rents lead to high incomes when political institutions are strong, but to corruption, violence and lower incomes when political institutions are weak, going back to [2,3], has been extremely influential. By now, many scholars, policy makers and international (donor) organizations believe that institutions are a key determinant of whether natural resource rents are a curse or a blessing. Given how influential this argument has become, one may expect that there is strong empirical evidence in support of it. [2,3] indeed provide some evidence, but rely on cross-country regressions like most of the literature of those times (e.g., [1,6–10]). Later research on the country-wide developmental effect of natural resources exploits intertemporal variation in: known resource deposits; actual resource extraction; or international commodity prices (e.g., [11–15]). Many recent contributions

**Data Availability Statement:** All replication files will be available on the corresponding author's website https://praschky.github.io/.

**Funding:** RH and PR: ARC Discovery Grant DP150100061, Australian Research Council,

https://www.arc.gov.au/ ML: SNSF
407540_166999, NRP 75 of the Swiss Science
Foundation, https://www.snf.ch/en The funders had
no role in study design, data collection and
analysis, decision to publish, or preparation of the
manuscript.

**Competing interests:** The authors have declared
that no competing interests exist.

exploit variation in local mining activities or the world market prices of locally mined minerals within countries across subnational units or within subnational units over time. Two prominent contributions are [16,17]. They study the effect of natural resources on conflict events and economic development in samples of subnational units from all over Sub-Saharan Africa. [18,19] provide a review of the recent literature focusing on the subnational level. The latter also discusses the remarkable increase in the availability of geo-spatial data that makes the focus on subnational units possible. While [16,17] focus on subnational units from all over Sub-Saharan Africa, other contributions focus on a single country (e.g., [20,21]) or a single mine (e.g., [22]). However, none of these more recent contributions focuses on the role of institutions.

In this paper, we reassess the hypothesis that institutions shape the effect of natural resource rents on economic development and political violence. We use a large sample of 3,800 districts (ADM2 regions) from all over Sub-Saharan Africa during the years 1996–2013. We divide the observations into a control group, consisting of all the districts that never had any mining activity during the sample period, and a few discrete treatment groups that differ by the relative prices of the locally mined minerals. We are interested in the effects of natural resources on economic development (measured by nighttime light emissions) and the occurrence of conflict events and, in particular, how these effects depend on institutional quality. Using a large and informative set of control variables, including various sorts of fixed effects, we base our identification on a selection-on-observable design. In such a set-up, we exploit recent developments in the causal machine learning literature that proposed very flexible, essentially nonparametric estimators to estimate causal effects at different aggregation levels without having to impose parametric modelling assumptions (as in a regression, for example). Additionally, these new estimators provide systematic ways to investigate the heterogeneity of the causal effects.

Concretely, we use the causal machine learning estimator by [23,24], which builds upon the one proposed by [25]. For our purpose, the main advantage of using this estimator is that we can estimate causal effects at different levels of aggregation: from the common average treatment effects (ATEs) to group specific treatment effects (GATEs) to very disaggregated individualized average treatment effects (IATEs). Therefore, this estimator is particularly suitable for systematically investigating effect heterogeneity. For example, by building groups based on standard indicators of institutional quality, the GATEs allow the study of effect heterogeneity along the institutional dimension.

There are some further important differences to the recent literature using subnational data: first, we use a non-parametric approach rather than a linear fixed-effects specification. We argue that our non-parametric approach is a worthwhile alternative because linear fixed-effects specifications are based on a set of assumptions that may not hold (see Section 2.1). Second, our approach offers a natural way to study the different types of (possibly heterogenous) "effects of natural resources" in a unified setting, namely the effects of mining activities per se as well as the effects of mining in times of low and high world market prices of the locally mined minerals. Finally, we use the same approach to estimate the (possibly heterogenous) effects of natural resources on economic development and conflict. This joint analysis is advantageous in that the differences in results cannot be driven by differences in the estimation approach used. There is ample theoretical work studying the effects of natural resource rents on economic development and conflict in a unified model (e.g., [5,26,27]). In a broader sense, rent-seeking activities may include political violence. Hence, the model of [2] could also be seen as a model of natural resource rents, economic development, and conflict.

We first study the ATEs. Our results suggest that mining activities increase local GDP by around 5 percent (based on an increase in nighttime lights by 16.0 percent) and the likelihood

of local conflict events by 7.9 percent if mineral prices are low. Similarly, high mineral prices (as opposed to low) increase local GDP and the likelihood of conflict events by around 10 and 6.5 percent, respectively. Moreover, these effects are non-linear. The effects of changes from low to intermediate prices are typically small and statistically insignificant, while the effects of changes from intermediate to high prices are typically much larger and statistically significant. We would not have detected this non-linearity if we had used a traditional linear fixed-effects specification. Hence, this finding represents a first example of the benefits of choosing a non-parametric and, therefore, non-linear estimation approach.

These ATEs, however, are based on IATEs that are quite dispersed. This dispersion underscores the importance of taking a closer look at effect heterogeneity. Thereby, we leverage another benefit of our estimator: the possibility of estimating GATEs. We focus on the role of institutions and group our observations by two prominent country-level measures of institutional quality: Constraints on the Executive from the Polity IV project and the Quality of Government from the International Country Risk Guide (ICRG) rating by the PRS Group. In support of the hypothesis by [2,3], we find that mining activities have more positive effects on economic development and weaker effects on conflict if institutional quality is high. Interestingly, however, we find that the effects of higher world market prices of locally mined minerals (rather than mining activities per se) do not vary systematically in institutional quality. That is, higher mineral prices have similar effects in both weakly and strongly institutionalized resource-extracting countries. This result implies that the hypothesis by [2,3] does not extend itself to the effects of changes in mineral prices. These findings are policy relevant. They suggest that well-meaning local policy makers and international (donor) organizations should indeed prioritize institutional safeguards for countries that start extracting natural resources, but that such safeguards may be less important in the case of increasing commodity prices.

The remainder of the paper is structured as follows. Section 2 discusses the key assumption underlying the estimation approach used in recent contributions, presents the causal machine learning estimator used in this paper, and discusses the identifying assumptions. Section 3 presents the data, and Section 4 the results. Section 5 briefly concludes.

## Econometrics and empirical approach

### Discussion of the recently used estimation approach

The recent literature using subnational data shows substantial concerns about possible selection effects coming from unobservables at the subnational unit level. These unobservables are jointly correlated with the local availability of resources and the outcome of interest, e.g., measures of economic development or the occurrence of conflicts. To address these concerns, most recent studies in this literature apply some form of two-way fixed effects model. For example, [16] include country-year- and cell-fixed effects, and [17] include year- and district-fixed effects. While these two-way fixed effects models are important improvements on the approaches used in the earlier literature, they still rest on restrictive assumptions.

Recently [28], point out that a key assumption of a two-way fixed effects model is that the treatment effect is constant across the individual geographic units and time. This assumption may well be violated for the effects of natural resource shocks on local economic development or the likelihood of local conflict. [28] show that in the presence of heterogenous treatment effects the estimated coefficients are biased. We investigate (and reject) effect homogeneity below. In addition, standard two-way fixed effects models rest on a linearity assumption. Of course, any model is wrong, but the nature of our dependent variables (the log of nighttime lights and a binary conflict variable) makes a linear relationship of the fixed effects, the other variables, and the resource variables unlikely, if not impossible.

Indeed, there is a very active literature on how standard differences-in-differences can be generalized. Examples of such papers are [29–32]. However, even after the various generalizations, many assumptions that are specific to differences-in-differences remain, like, for example, the measurement dependence of the common trend assumption which allows differences to take place (Due to the fact that, for example, the difference of a log does not equal the log of a difference.).

Therefore, we explore an alternative approach suggested by the recent developments in causal machine learning. This alternative has many advantages, one of which is that it does not depend on the functional specification (unit of measurement) of the outcome variable, which we discuss below. However, it comes at the cost of not being able to include that many fixed effects (for details on the variables included, see Section 3). Hence, there is a trade-off. We consider our approach a worthwhile alternative or complement to the approaches used in the literature so far.

## The modified causal forest estimator

In this paper, we utilize the recently upcoming causal machine learning literature (see, e.g., [33]). It combines the prediction power of the machine and statistical learning literature (see, e.g., [34], for an overview) with the microeconometric literature on defining and identifying causal effects (see, e.g., [35]). Recently, this literature has seen a surge of proposed methods that emphasize the reliable detection of effect heterogeneity (see, e.g., [36] for an overview). Here, we use a promising candidate of these estimators, the Modified Causal Forest (MCF) estimator proposed by [23].

The starting point of the causal forest literature is the Causal Tree introduced by [37]. In a Causal Tree, the sample is split sequentially into smaller and smaller strata in which the values of X become increasingly homogenous, to mitigate selection effects and to uncover effect heterogeneity. The treatment effect is computed within each stratum (called a "leaf") as the difference of the mean outcomes of treated and controls (possibly weighted by the conditional-on-X probabilities of being a treated or control observation). However, the resulting final leaves may be rather unstable because of the sequential nature of the splits. Random Forests are a solution to this problem. They induce some randomness into the tree building process, build many trees, and then average the predictions of the many trees. [25] use this idea to propose their Causal Forests, which are based on a collection of Causal Trees with small final leaves. [38] generalize this idea to many different econometric estimation problems. [23] develops these ideas further by improving on the splitting rule for the individual trees. This improvement is done by finding an alternative way of minimizing the mean squared error of the estimated IATEs which involves some intermediate matching step that leads to improved performance for the resulting MCF. Furthermore, the MCF allows for the estimation of the ATE and heterogeneous effects for a limited number of discrete policy variables (i.e., GATEs) at low computational costs, in addition to the highly disaggregated IATEs on which the previous literature focused. Furthermore, the MCF provides unified inference for all aggregation levels. The main idea is to estimate the effects at the lowest aggregation level (i.e., the IATE) and then aggregate those to obtain GATEs and ATEs. This aggregation is attractive because predictions from Regression Random Forests and Causal Forests can be represented as weighted means of the outcomes, where the weights can directly be obtained from estimated forest. Thus, only the weights need to be aggregated. This is also useful because inference for all aggregation levels is based on exactly those weights.

Finally, the MCF is applicable to a multiple, discrete treatment framework. We use the MCF because these advantages are important for our empirical analysis. [23] presents further technical details of the MCF estimator.

## Identification

To estimate the IATEs, GATEs and AETs in a non-parametric way, we discretize the treatment and divide the observations (district-years) into *M* treatments. We do so in two steps. First, we distinguish between mining and non-mining districts, and refer to the latter as control group (or Treatment 0 group). Second, we allocate all the observations from the mining districts into *M-1* approximately equally sized treatment groups depending on a district-specific mineral price indicator (see Section 3.2 for details). Hence, in case of *M = 3*, we separate the observations of mining districts into those with relatively low and high prices of the minerals mined in that area. In the case of *M>3*, we can investigate potentially non-linear effects. In any case, we estimate the average effects of the treatments that fall into the respective price indicator brackets. Such a fully non-parametric approach has the advantage that it cannot be subject to a misspecification of functional forms.

Along with the recent subnational literature, we argue that the availability of natural resources at the district level is mainly shaped by geology and, therefore, exogenous. This argument is certainly true for the actual presence of natural resources. However, it is important to acknowledge that it may be endogenous whether these resources are discovered and exploited (e.g., [39]). We also follow the recent subnational literature in arguing that the world market prices are also exogenous to any single district as long as we focus on natural resources extracted in many places.

Let us now discuss the identifying assumptions in more detail. The classical set of unconfoundedness assumptions consists of four parts: conditional independence, common support, stable-unit-treatment value, and exogeneity. [40,41] discuss these assumptions and their identifying power concerning various average causal effects (like the ATEs). Those results are also sufficient to nonparametrically identify the IATEs and GATEs (e.g., [28]).

The conditional independence assumption (CIA) requires that no variables other than the observable *X* jointly influence the treatment and the potential outcomes. Here, the plausibility of CIA is enhanced by the fact that we assemble data for a large set of potential confounders that have been identified in the literature, and that we control for the confounding and heterogeneity variables non-parametrically and, therefore, much more flexibly than we could in linear specifications. Firstly, a large theoretical and empirical literature has highlighted the interplay between: natural resources; institutional quality; and economic development (or conflict) (e.g., [2,3,5]). We have therefore compiled data on national and subnational institutional quality (beyond the measures used as heterogeneity variables). Secondly, ever since the seminal work by [1,6], it has been fairly standard in the empirical resource curse literature to account for differences in the geographic and climatic factors between the units of analysis. For example, more recent empirical work by [42] highlights the importance of geography for explaining differences in economic development at the subnational level. It is also likely that the location of mineral deposits is correlated with topographic features of an area, such as ruggedness or soil characteristics, which in turn can have an effect on the area's long-term economic development as well as prevalence of violent conflict (e.g., [43,44]). Variations in climate conditions, in turn, not only have a direct impact on a subnational region's economic development but also influence the opportunity costs of fighting (e.g., [45]). Thirdly, we include control variables for subnational levels of population and population density, as natural resource wealth may attract more people and may alter local settlement patterns in the long-run. Relatedly, we also control for country- and local-level measures of ethnic diversity. Lastly, we include proxies for the transportation infrastructure. The reason is that a lot of the transportation infrastructure in Africa is mining-related (e.g., [46]), which can create positive spillovers for local economic development (e.g., [47]) or spread local conflict (e.g., [48]). Indeed, adding a large set of

covariates is no panacea for a potential omitted variable bias through other, unobserved confounding variables. For example, our measures of local infrastructure only partially picks up local variation in public goods provision and local differences in educational attainment, state capacity or other infrastructure can influence both mining activity as well as economic development and conflict likelihood. In general, these type of omitted variables can lead to a positive bias and our results would overestimate the true effect.

The common support assumption requires that for all values of the confounding and heterogeneity variables, every treatment state has a positive probability of getting selected. This assumption is needed as the nonparametric identification idea is based on comparing observations with similar covariate values across treatment states. If this assumption is violated, identification will depend on functional form (or similar) assumptions that (implicitly) allow to impute such comparison observation from the econometric model. We investigated this assumption by using Random Forest based estimates of the conditional probabilities to end up in the different treatment states and found substantial overlap problems. Since the literature has so far not considered this issue, we decided for reasons of comparability, to do the same. Practically, the MCF estimator is still feasible. In each tree, it will simply stop splitting further when the next split would lead to a leaf in which one or more treatments are missing. In other words, in a different but conceptually similar way to a regression-type model, it will extrapolate the required conditional outcome expectations into the regions without common support.

The final two assumptions are not specific to our approach but are needed by the standard approaches of the current literature as well. The stable-unit-treatment value assumption (STUVA) requires that the observed value of the treatment does not depend on the treatment allocation of the other units. STUVA requires the absence of spillovers. The evidence on spillovers is mixed: [17] find no evidence for spillovers from mining, while [47] find evidence for spillovers from changes in mineral prices among districts connected by roads or ethnolinguistic linkages.

The exogeneity assumption requires that the observed values of the confounding and heterogeneity variables do not depend on the treatment status. The plausibility of this assumption is enhanced by focusing on the following two sets of heterogeneity and control variables: predetermined variables that measure a district's geography or history, and country-level variables that should only marginally depend on the treatment status of a single district, given the large number of districts per country. In addition, we lag the treatment, heterogeneity, and control variables by one year.

## Data

Following [17], we focus on subnational administrative units. Our sample consists of 3,800 districts from 42 Sub-Saharan African countries. Our definition of districts is based on ADM2 regions according to the database of Global Administrative Areas (GADM) (Version 1). We use the ADM2 shapefiles by GADM to compute the various district-level variables based on geospatial data. Choosing the ADM2 federal unit as our preferred geographic unit has a number of advantages over other types of spatial resolution. Moving to an even lower federal level, such as ADM3 or small rectangular grid cells, would likely add additional noise to the data. For one of our outcome variables, nighttime light, there is the concern that the correlation between luminosity and economic activity may decrease for very small geographic units (see [49] and Section 3.1). Georeferenced data, such as the boundaries of ethnic homelands, is less precise and the likelihood that an ethnic homeland is not assigned to the correct geographic unit increases as the level of disaggregation becomes smaller. In contrast, using more aggregate geographic units, such as ADM1 (states), creates the problem that the unit becomes so large

that one may not identify the economic effects from very localized resource shocks. Our sample period covers 17 year, lasting from 1997 to 2013 for our outcome variables and from 1996 to 2012 for our treatment, heterogeneity, and control variables. We focus on the years 1997–2013, because one outcome variable is only available from 1997 onwards and the other only up to 2013. Conflict data based on Armed Conflict Location Events Data (ACLED) only start in 1997 while nighttime luminosity data from the US Defense Meteorological Satellite Program (DMSP) satellites end in 2013.

In Sections 3.1–3.3 we introduce our outcome, treatment and heterogeneity variables (and in Section 3.4 the confounding variables). As we aim to investigate what new insights we can gain from using a novel estimator (the modified random forest estimator), we deliberately use variables that have been used in the previous literature. Otherwise, it would be difficult to know whether different results are due to the use of different variables or the use of the novel estimator.

## Outcome variables

We follow a large and growing literature in using nighttime light intensity to proxy for economic development at the subnational level and in particular the recent work using nighttime light intensity to study the effect of mining activities on economic development at the subnational level include (e.g. [17,47]). These data are based on daily measures from the Operation Linescan System of the DMSP. The National Oceanic and Atmospheric Administration (NOAA) processes these measures with the objective of reporting only man-made nighttime light emissions and provides annual data for the period from 1992–2013 for output pixels of less than one square kilometer. The data come on a scale from 0 to 63, with higher values implying more intense nighttime lights. Nighttime lights are a proxy for economic activity, as many forms of consumption and production in the evening require light. Moreover, public infrastructure is often lit at night. [50,51] indeed find a high correlation between changes in nighttime lights and GDP at the level of countries and provinces, respectively. Complementarily, [52] document a positive relation between nighttime lights and broader measures of human development at the local level. As an outcome variable, we use the natural logarithm of the average nighttime light pixel value in a given district and year. To avoid losing observations with zero reported nighttime lights, many scholars have added 0.01 before taking logs (e.g., [17,47,51,53]). We use this transformation as well in order to ensure the comparability of our results to those in the previous literature. However, this transformation may not be innocuous. [54] argue in a recent working paper that the estimated ATEs may be scale dependent. Given the absence of any other data that provides a comparable, consistent and granular measure of subnational economic development across the African continent, nighttime luminosity is the only proxy to conduct an empirical study that exploits both spatial and temporal variation at fine granularity and requires consistently measured data for a comparative institutional analysis.

Despite these advantages, it is important to remember that DMSP satellite data on nighttime light only partially indicates economic activity being undertaken in an area. Using nighttime lights as a proxy for local development has shortcomings, especially in least developed countries and their thinly inhabited, rural areas and generally tiny geographical units (see, e.g., [55,56]). In particular, the sensors of the older generation DMSP satellites are unable to pick up lower levels of economic activity that emit no or only very weak luminosity signals. This so-called "bottom coding" problem is particularly a problem in very poor areas but [56] also showed that this issue can occur in more populated areas and areas with more economic activity. [49] combines more detailed income figures from Indonesia with DMSP satellite data and

found that the correlation between the two variables becomes weaker as the size of the geographic unit decreases. This is especially problematic if luminosity data are employed cross-sectionally to explain differences in development between subnational divisions. These results have the following implications for our setting. Due to the bottom coding problem, we are only able to pick up larger effects of changes in local mineral wealth on local economic activity suggesting that our estimates present a lower bound of the true effect. We think that the issues related to weaker correlation between nighttime light and economic activity in very small geographic units (i.e., grid cells) only partially affects our study, because our unit of analysis is at the ADM2 level, which tends to be moderately large. A number of recent studies (e.g., [49,56,57]) recommend the use of nighttime luminosity data produced by the more recent generation of satellites, the Visible Infrared Imaging Radiometer Suite (VIIRS). However, the luminosity data series from VIIRS only starts in 2013 and the results time series would be too short for our empirical strategy which relies on longer time-series variation in world mineral prices.

To construct our outcome variable for conflict, we rely on two prominent and frequently used geo-referenced datasets on conflict events: The ACLED and the UCDP Georeferenced Event Dataset. These datasets cover conflict events starting in 1997 and 1989, respectively. Our primary conflict outcome is a binary variable indicating whether at least one conflict event took place in a given district and year according to at least one of these two datasets. The share of district-years with a conflict event is 13.4 percent. This variable is like the main dependent variable in [16], with the only difference being that we construct a single variable based on both datasets in order to get a somewhat lower share of zeros. By transforming the raw conflict data into an aggregate, binary outcome variable, we abstract from the heterogeneity of conflicts along intensity, duration or type (i.e., battles, violence against civilians, etc.). A drawback of this aggregation is that increases in local resource rents could potentially have differential, or even opposing, effects on the type of conflict. Our approach only allows for the estimation of a net effect on local conflict. Similarly, this study only focuses on the extensive margin of the effect of resource wealth on conflict and not the intensive margin (e.g., intensity and duration). The purpose of this binarization is to use an outcome variable similar to the ones used in existing studies that estimate the impact of resource rents on conflict (e.g., [16]) to compare results based on different estimation approaches. Needless to say, a more detailed study on the effects of exogenous resource wealth shocks on local conflict type, duration and intensity would be of interest, but a careful analysis of these questions would go beyond the scope of this paper.

## Treatment variables

We divide our observations into a control group, consisting of all districts without any mining activity, and a discrete number of treatment groups that differ by the relative prices of the locally mined minerals. For this purpose, we use information on mining activity from the SNL Metals & Mining database and information on the world market prices of minerals from various sources (see Table A1 in Online Appendix in S1 Appendix). Hence, we rely on the same type of data for natural resources as [16,47].

In a first step, we use the SNL Metals & Mining database to assign districts without mining activity to the control (or Treatment 0) group. To do so, we keep all 2,326 mineral mining projects across Sub-Saharan Africa that were producing minerals in at least one year during our sample period. For each project, this database contains information on the geo-coordinates and the (potentially multiple) minerals extracted at this location. We use the geo-coordinates to assign the mining projects to districts. The share of districts that contain at least one mine is 6.0 percent. Below we explain how we assign each of these mining districts to a treatment

group. The share of districts without a mine is 94.0 percent. These latter districts constitute our control group. The evidence on spillover effects of natural resource rents on economic development or conflict in neighbouring areas is inconclusive. For example, [17] (p. 28) find "little evidence for significant [economic] spillovers to other districts," while [47] provide evidence for economic spillovers between districts that are well connected by roads or part of the same ethnic homeland. Given that we assign non-mining districts that are contiguous to mining districts to the control group, the presence of spatial spillovers would lead to attenuation bias.

In a next step, we assign each mining district to a discrete number (*M-1*) of treatment groups. We proceed as follows. First, we divide the annual world market price of each mineral by its mean value over the sample period (normalization). Second, for each mining district, we identify the set of minerals extracted in this district during our sample period and average the normalized world market prices across these minerals. This average is our district-specific mineral price indicator. Third, we define brackets for the district-specific mineral price indicator that ensure that we assign the mining regions to *M-1* treatment groups with the same number of observations.

## Heterogeneity variables

A key contribution of this paper is to present GATEs to improve our understanding of effect heterogeneity and, in particular, the role of institutions. For this purpose, we want to select two measures of country-level institutional quality as heterogeneity variables *Z*. There exist many indicators of institutional quality, often measuring different aspects of institutional quality or measuring the same aspect with different methodological approaches. The Quality of Government Institute at the University of Gothenburg provides a compilation dataset of indicators related to institutional quality. Among others, they list more than 100 indicators in each of the following categories: "Judicial", "Political System", and "Quality of Government." At a general level, indicators related to institutional quality can differ because of differences in the conceptualization of quality of government, the operationalization of the components of this concept, and the aggregation rule used to create a unidimensional index ([58,59]). As all indicators have their strengths and weaknesses, there exists no obvious way of determining the best indicators. We thus use the following two selection criteria. First, following our general strategy mentioned above, we choose indicators that have been commonly used in the previous literature in order to ensure the comparability of our results to those in the literature. Second, [60] and others argue that the quality of a country's economic institutions may be endogenous to the quality of its political institutions. Therefore, following a this general distinction in the literature, we want an indicator that mainly captures the quality of the political institutions that constrain the government, and another one that mainly captures the quality of the economic institutions that shape the economic incentives of private individuals and firms.

The first indicator is the one that [2,3] used in their original contributions: the Quality of Government indicator from the ICRG rating by the PRS Group. This indicator corresponds to the average of three indices measuring: corruption; law and order; and bureaucratic efficiency. It ranges from 0–1, with higher values implying higher quality. As it measures the government's performance and effectiveness, it is best seen as an indicator of the quality of the economic institutions that shape private investment decisions.

The second institutional variable is the Constraints on the Executive score from the Polity IV project, which measures the extent of institutionalized constraints on the decision-making powers of the chief executive. Effective constraints on those with decision-making powers are generally seen as an important (or even the most important) characteristic of sound political institutions ([60]). The Constraints on the Executive score ranges from 1–7, with higher values

implying more constraints and, therefore, better political institutions. This and other measures from the Polity IV project have been studied extensively in various strands of the literature in economics and political science, including the closely related paper by [16]. On a global scale, the best countries to illustrate the different traits captured by these two institutional variables are Singapore and Paraguay. Singapore is not very democratic, but has an effective and reasonably benevolent government, which is reflected by a low score for constraints on the executive (3.00, averaged over the sample period) but a high score for the quality of government (0.87). Paraguay, in contrast, is a democratic country with poor governance, which is reflected by a high score for constraints on the executive (6.94) but a low score for the quality of government (0.31). Focusing on Sub-Saharan Africa, the comparison between Gambia and Liberia reveals a similar, albeit weaker, pattern. Gambia has low constraints on the executive (1.94) but relatively high quality of government (0.56), while Liberia has relatively strong constraints on the executive (4.15) but low quality of government (0.23). The correlation between these two institutional variables across Sub-Saharan African countries is 0.04.

## Confounding variables

As discussed in Section 2.3, we include many potentially confounding variables $X$ (beyond the $Z$ variables introduced in Section 3.3). These variables can be divided into two sets. The first consists of variables capturing variation at the subnational level while the second set consists of variables which vary at the country and year level. It contains indicator variables for countries, years, and country-years. A more detailed description of how each of the confounding variables was calculated, as well as the sources of the underlying raw data, can be found in the online appendix.

The district-level variables are mostly based on geo-spatial data and computed using the ADM2 boundaries from the GADM database of Global Administrative Areas (Version 1).

Land *Area* measures the ADM2's area in km$^2$. *Distance to Capital* and *Distance to the Coast* are the log of the distance between the district's geographic center, the country's capital in km and the nearest coast in km, respectively. *Elevation* refers to each district's minimum/median/maximum elevation, respectively. *Land Use* is a set of 11 variables each representing the share of the land area of 11 different types of land cover (i.e., artificial, crop, grass, trees etc.). *Land suitability for agriculture*: is the district's average land suitability for the cultivation of crops based on climate and soil constraints. *Ruggedness* is based on the average absolute height difference between a raster pixel and its neighbors to calculate the Mean Terrain Roughness Index. *Temperature and precipitation* are the annual mean temperature and precipitation values for each district.

In addition to these geographic variables, we also calculated a set of socio-economic and historical variables at the district level. *Number of traditional ethnic homelands* is the sum of ethnic homelands (based on georeferenced boundaries of these homelands) with the boundaries of a given ADM2 district. *Pre-colonial centralization* describes the degree of pre-colonial, political centralization (i.e., villages, state-like) in each of the traditional ethnic homelands. *Population and Population Density* are the log of the number of people and people per km$^2$ in each district, respectively. *Ports (with and without oil terminals)* is a dummy variable that switches to one if a port (a port-with-oil-terminal) is located in the district, and zero otherwise.

The second set of confounding variables vary at the country and year level. This set contains countries, years, country-years, indicators for the former colonial rulers, the indices of ethnic and religious fractionalization, GDP per capita (in constant 2010 USD and as annual growth rate) and inflation, and a large set of institutional variables: the indices of Civil Liberties and Political Rights by Freedom House, the Polity2 score by the Polity IV Project, the six World-wide Government Indicators (i.e., Control of Corruption, Government Effectiveness, Political

Stability, Regulatory Quality, Rule of Law, Voice and Accountability, whether the political system is presidential or parliamentary, and whether the political system is unitary or federal). Table 1 presents the descriptive statistics of the main variables.

# Results

## Average effects

Table 2 presents the ATEs for our two outcome variables and the different treatment variables. Panel A focuses on the case of $M = 3$, i.e., the case in which the district-years of mining districts are discretized into two treatments, characterized by relatively low and high current world market prices of the locally mined minerals. The columns on the left present results for our measure of local economic development, the log of nighttime lights. The first two rows present the estimated effects of the low-price treatment (Treatment 1) and the high-price treatment (Treatment 2), as compared to the control group of districts without mining activities (Treatment 0). The third row compares Treatments 1 and 2. We find a statistically significant, positive effect of mining activities on economic development for both low and high mineral prices. This effect is much stronger in the case of high prices, with the difference being statistically significant as well. In particular, the estimated effects imply that mining activities increase nighttime lights by 16.0 percent if mineral prices are low, and 48.3 percent if mineral prices are high. In comparison, [17] find that nighttime lights increase by 55 percent in districts that start industrial mining. Hence, this effect is similar to the effect we get in the case of high mineral prices.

The following question arises: what do our estimated increases in nighttime lights imply for economic development arises? [50,51] report elasticities between nighttime lights and GDP of around 0.3 at the level of countries and provinces, respectively. Assuming the same elasticity at the district level, our estimated effects imply that mining activities increase local GDP by around five percent if mineral prices are low, and around 15 percent if mineral prices are high.

The columns in the right of Table 3 use our binary conflict variable as the outcome. We find a similar pattern in the sense that mining activities increase the likelihood of conflict for both low and high mineral prices, but that this effect too is stronger for high prices. More precisely, the likelihood of conflict is 7.9 percentage points higher in mining districts than in non-mining districts if prices are low, and 14.4 percentage points higher if prices are high. Hence, in mining districts, the likelihood of conflict is 6.5 percentage points higher if the price of the locally mined minerals is high rather than low. This difference is roughly comparable to [16] finding that a one-standard deviation increase in the price of locally mined minerals raises the likelihood of conflict in mining grid cells by 5.6 percentage points.

Panel B reports results for the case of $M = 4$, in which the observations from mining districts are discretized into three treatments representing low, intermediate, and high prices of the locally mined minerals. The results suggest non-linear effects of mineral prices on economic development and conflict. For nighttime lights, the difference between the high- and the intermediate-price treatment (comparison 3–2) is more than four times larger than the difference between the intermediate- and the low-price treatment (comparison 2–1). For conflict, the former difference is basically zero, while the latter difference is again substantial. Panel C, which reports results for the case of $M = 5$, also suggests non-linear effects of mineral prices on economic development and conflict.

## Effect heterogeneity and the role of institutions

Remember that the ATEs presented above, and the GATEs discussed below, are computed based on the estimated IATEs. Fig 1 illustrates the extent of the deviations of the IATEs from

**Table 1. Descriptive statistics.**

| Variable | Mean | Std. Dev. | Min | Max |
|---|---:|---:|---:|---:|
| | | District Level | | |
| Conflict | 0.14 | 0.35 | 0.00 | 1.00 |
| Nighttime Light | 1.57 | 6.03 | 0.00 | 63.00 |
| **Sum Resources** | 0.10 | 0.52 | 0.00 | 7.00 |
| Mineral price indicator | 0.06 | 0.27 | 0.00 | 3.02 |
| Area (km$^2$) | 4449.55 | 11423.19 | 4.18 | 224667.10 |
| Ruggedness | 25.07 | 33.21 | 0.00 | 283.47 |
| Elevation (min, meters) | 488.13 | 452.45 | -169.00 | 2087.00 |
| Elevation (max, meters) | 1067.18 | 820.97 | 20.00 | 5825.00 |
| Elevation (median, meters) | 675.61 | 540.62 | 1.00 | 2879.00 |
| Distance to Capital (meters) | 309970.50 | 265009.90 | 10231.52 | 1825133.00 |
| Land Use—Artificial | 0.01 | 0.05 | 0.00 | 0.97 |
| Land Use—Cropland | 0.30 | 0.25 | 0.00 | 1.00 |
| Land Use—Grass | 0.21 | 0.25 | 0.00 | 1.00 |
| Land Use—Trees | 0.22 | 0.22 | 0.00 | 0.99 |
| Land Use—Shrubs | 0.19 | 0.18 | 0.00 | 0.99 |
| Land Use—Herbaceous | 0.01 | 0.04 | 0.00 | 0.85 |
| Land Use—Mangroves | 0.00 | 0.03 | 0.00 | 0.61 |
| Land Use—Sparse | 0.01 | 0.06 | 0.00 | 0.90 |
| Land Use—Baresoil | 0.03 | 0.13 | 0.00 | 1.00 |
| Land Use—Snow | 0.00 | 0.00 | 0.00 | 0.00 |
| Land Use—Water | 0.02 | 0.08 | 0.00 | 1.00 |
| Temperature (mean, deg. Celsius) | 24.33 | 3.98 | 5.48 | 32.11 |
| Precipitation (mm) | 87.58 | 44.47 | 0.10 | 357.18 |
| Agricultural Suitability | 0.37 | 0.23 | 0.00 | 1.00 |
| Ethnic Groups (Ethnologue) | 7.38 | 8.54 | 2.00 | 41.00 |
| Ethnic Groups (Murdock) | 3.84 | 2.96 | 2.00 | 19.00 |
| Pre-Colonial Polity | 0.66 | 0.47 | 0.00 | 1.00 |
| Ports | 0.04 | 0.23 | 0.00 | 4.00 |
| Ports—Oil Terminal | 0.01 | 0.13 | 0.00 | 2.00 |
| Population (interpolated) | 164552.70 | 278605.80 | 159.40 | 5838058.00 |
| Population Density | 224.59 | 795.86 | 0.01 | 19821.45 |
| | | Country Level | | |
| ICRG Quality of Government | 0.40 | 0.10 | 0.08 | 0.88 |
| Executive Constraints | 8.73 | 4.50 | 2.00 | 17.00 |
| WGI Control of Corruption | -0.62 | 0.59 | -1.70 | 1.22 |
| WGI Government Effectiveness | -0.63 | 0.57 | -2.20 | 1.02 |
| WGI Political Stability | -0.71 | 0.84 | -2.86 | 1.20 |
| WGI Rule of Law | -0.71 | 0.56 | -2.42 | 0.73 |
| WGI Regulatory Quality | -0.54 | 0.59 | -2.25 | 0.80 |
| WGI Voice and Accountability | -0.50 | 0.64 | -2.23 | 0.86 |
| Polity2 | 2.20 | 4.62 | -7.00 | 9.00 |
| IAEP Unitary or Federal State | 1.52 | 0.87 | 1.00 | 3.00 |
| GDP p.c. (2010, PPP) | 1770.75 | 2103.85 | 186.66 | 20333.94 |
| Freedom House Political Rights | 4.13 | 1.66 | 1.00 | 7.00 |
| Freedom House Civil Liberties | 3.96 | 1.24 | 2.00 | 7.00 |
| Ethnic Fractionalisation | 0.74 | 0.14 | 0.26 | 0.93 |

*(Continued)*

**Table 1.** (Continued)

| Variable | Mean | Std. Dev. | Min | Max |
|---|---|---|---|---|
| Religious Fractionalisation | 0.66 | 0.17 | 0.00 | 0.86 |
| Colonial Origin | 4.25 | 2.13 | 2.00 | 10.00 |

Note: Descriptive Statistics. No. of obs. 60121.

the ATEs by presenting sorted IATEs on nighttime lights for $M = 4$. To economize on space, we restrict our attention to the effects of a low-price treatment as compared to no treatment (comparison 1–0, left graph) and a high-price treatment compared to no treatment (comparison 3–0, right graph). In both comparisons, the deviations of the IATEs from the ATEs vary from around -0.6 to around 0.4. The corresponding deviations are also sizeable in case of conflict. In addition, we further supplement the results by showing the feature importance index for NTL with 3 categories. Table 4 presents the variable importance statistics in terms of %-lost of the base value for the most important variables.

This heterogeneity in the IATEs implies that there is considerable room for potential effect heterogeneity along the dimension of the two institutional measures introduced in Section 3.3. Therefore, we now turn to the corresponding GATEs. We thereby again focus on

**Table 2.** Average effects for different outcome and treatment variables.

| Comparison | Log nighttime lights | | | Conflict (0/1) | | |
|---|---|---|---|---|---|---|
| | Effect | Std. | p-value in % | Effect | Std. | p-value in % |
| | Panel A: $M = 3$ | | | | | |
| 1–0 | 0.148 | 0.050 | 0.3 | 0.079 | 0.022 | 0.0 |
| 2–0 | 0.394 | 0.045 | 0.0 | 0.144 | 0.022 | 0.0 |
| 2–1 | 0.248 | 0.067 | 0.0 | 0.065 | 0.031 | 3.7 |
| | Panel B: $M = 4$ | | | | | |
| 1–0 | 0.195 | 0.059 | 0.1 | 0.094 | 0.027 | 0.1 |
| 2–0 | 0.248 | 0.054 | 0.0 | 0.092 | 0.024 | 0.0 |
| 3–0 | 0.473 | 0.050 | 0.0 | 0.152 | 0.030 | 0.0 |
| 2–1 | 0.053 | 0.078 | 50.4 | -0.002 | 0.036 | 94.3 |
| 3–1 | 0.278 | 0.077 | 0.0 | 0.058 | 0.040 | 15.2 |
| 3–2 | 0.225 | 0.073 | 0.2 | 0.060 | 0.038 | 11.2 |
| | Panel C: $M = 5$ | | | | | |
| 1–0 | 0.247 | 0.062 | 0.0 | 0.119 | 0.038 | 0.2 |
| 2–0 | 0.220 | 0.065 | 0.1 | 0.098 | 0.036 | 0.7 |
| 3–0 | 0.352 | 0.054 | 0.0 | 0.150 | 0.031 | 0.0 |
| 4–0 | 0.574 | 0.053 | 0.0 | 0.215 | 0.038 | 0.0 |
| 2–1 | -0.027 | 0.090 | 76.5 | -0.213 | 0.053 | 68.7 |
| 3–1 | 0.105 | 0.081 | 19.6 | 0.039 | 0.049 | 41.7 |
| 4–1 | 0.327 | 0.081 | 0.0 | 0.010 | 0.054 | 7.6 |
| 3–2 | 0.132 | 0.084 | 11.5 | 0.061 | 0.047 | 20.0 |
| 4–2 | 0.354 | 0.083 | 0.0 | 0.117 | 0.052 | 2.7 |
| 4–3 | 0.222 | 0.075 | 0.3 | 0.056 | 0.049 | 25.0 |

Note: Top row indicates outcome variable. Left-most column indicates comparison between different treatment group. Treatment group 0 consists of all non-mining districts. The other treatment groups of mining districts are divided into $M$-1 groups with the same number of observations, with higher treatments representing higher prices (see Section 3.2 for details). Standard errors account for clustering due to the panel structure at the district level.

**Table 3. Summary of relevance of GATEs for all prespecified variables: Wald tests.**

| Comparison | Log nighttime lights | | | Conflict (0/1) | | |
|---|---|---|---|---|---|---|
| | Chi2 | df | p-value in % | Chi2 | df | p-value in % |
| | Panel A: Constraints on the executive | | | | | |
| 1–0 | 43 | 7 | 0 | 41 | 7 | 0 |
| 2–0 | 59 | | 0 | 31 | | 0 |
| 3–0 | 127 | | 0 | 18 | | 1 |
| 2–1 | 2 | | 92 | 2 | | 87 |
| 3–1 | 12 | | 7 | 4 | | 74 |
| 3–2 | 6 | | 40 | 3 | | 85 |
| | Panel B: Quality of government | | | | | |
| 1–0 | 62 | 18 | 0 | 89 | 18 | 0 |
| 2–0 | 56 | | 0 | 64 | | 0 |
| 3–0 | 99 | | 0 | 49 | | 0 |
| 2–1 | 6 | | 99 | 9 | | 95 |
| 3–1 | 41 | | 0 | 31 | | 3 |
| 3–2 | 19 | | 37 | 24 | | 15 |

Note: Standard errors account for clustering due to panel structure.

specifications with three treatment groups, i.e., mining districts with low, intermediate and high mineral prices, and the non-mining districts as control groups ($M = 4$). Table 3 shows the Wald tests for heterogeneity for the GATEs of each of our heterogeneity variables. The null hypothesis is that all effects are the same. A large test statistic together with a small p-value thus indicates an increased likelihood of heterogeneity with respect to that specific variable. Observations are grouped based on the Constraints on the Executive score in Panel A and the Quality of Government indicator in Panel B. We stratify the (continuous) Quality of Government indicator into 19 groups of approximately equal size. This number appears as a reasonable compromise between precision (fewer groups) and more heterogeneity (more groups). The Wald tests for the comparisons 1–0, 2–0 and 3–0 strongly suggest that the effects of mining activities on nighttime lights and conflict depend on institutional quality, regardless as to whether the mineral prices are low, intermediate or high. The relation between institutional quality and the effects of changes in mineral prices are more nuanced. The effects of moderate price changes (i.e., comparisons 2–1 and 3–2) do not depend on institutional quality in a

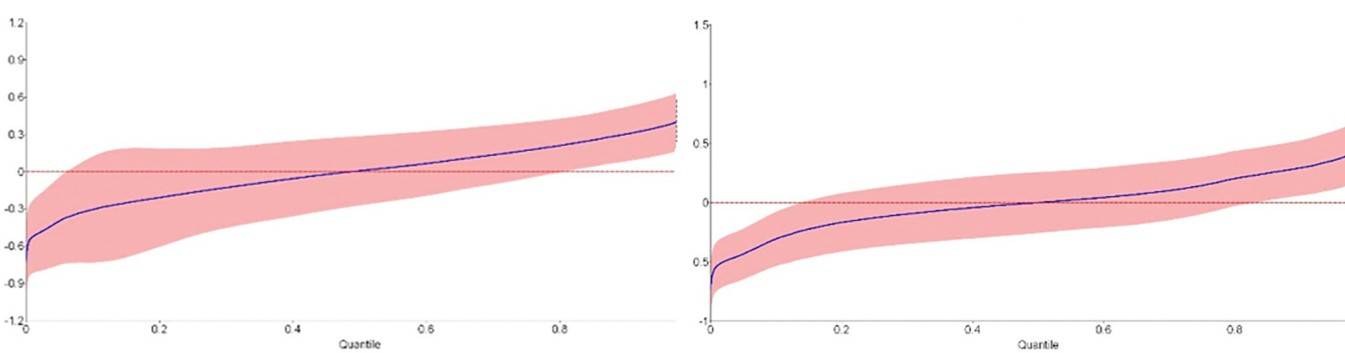

**Fig 1. Sorted IATEs on nighttime light intensity relative to the ATEs (comparisons 1–0 and 3–0, M = 4).** Note: The shaded area shows the 90% confidence interval of the difference of the IATE and the ATE. Horizontal axes: Quantiles of the ordered effects. Vertical access: Values of (IATE–ATE).

**Table 4. Variable importance statistics in %-lost of base value (NTL).**

| Variable | %-lost |
|---|---|
| WBGI_PVE | 47.14% |
| WBGI_PVECATV | 47.14% |
| ADM1ID | 27.64% |
| TEMPERATURE_MEAN | 22.20% |
| FRAC_LU_GRASS | 17.53% |
| WDI_INFLATION | 14.13% |
| WBGI_GEE | 13.92% |
| WBGI_GEECATV | 13.92% |
| PRE_COL_POLITY | 12.27% |

statistically significant manner, while the effects of large price changes (i.e., comparison 3–1) do, especially when using the Quality of Government indicator.

Figs 2 and 3 present the GATEs to illustrate the specific heterogeneity for these indicators of institutional quality. To economize on space, we again focus on the 1–0 and 3–1 comparisons. The 1–0 comparison shows that the GATEs of mining activities (at low mineral prices) on economic development are largest if institutional quality is high and lowest if institutional quality is low, while the reverse pattern holds for the GATEs of mining activities on conflict. These findings support the hypothesis by [2,3] and the general argument that institutions shape the effect of resource extraction on economic development and conflict. The 3–1 comparison in Figs 2 and 3 suggest that the GATEs of higher (as opposed to lower) mineral prices

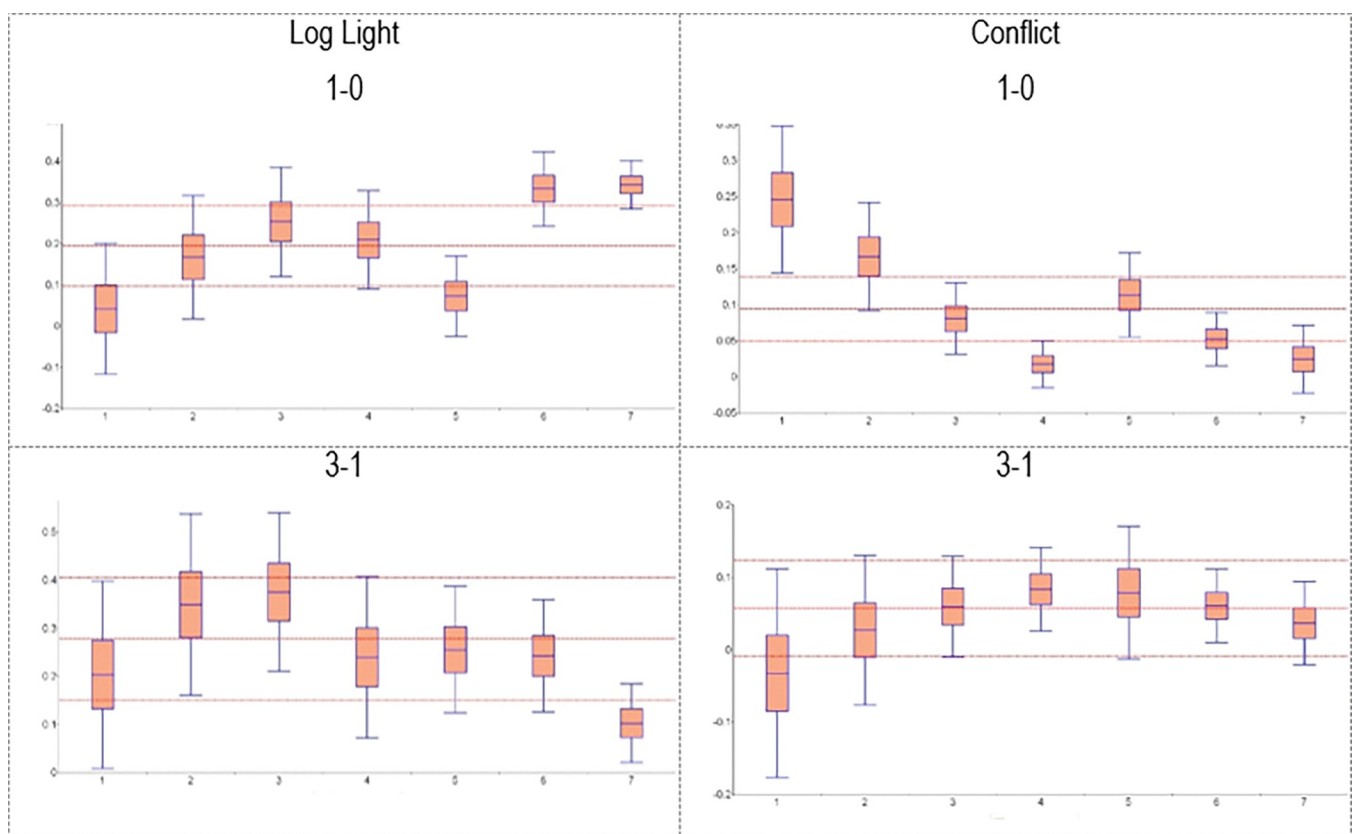

**Fig 2. GATEs for different levels of constraints on the executive (M = 4).** Note: The graphs show the GATEs for the various levels as well as their 90% confidence interval. The horizontal red lines indicate the ATEs and their 90% confidence interval (reported in Panel B of Table 1). Horizontal axes: Values of the variable constraints on the executive. Vertical access: Values of GATEs and ATE.

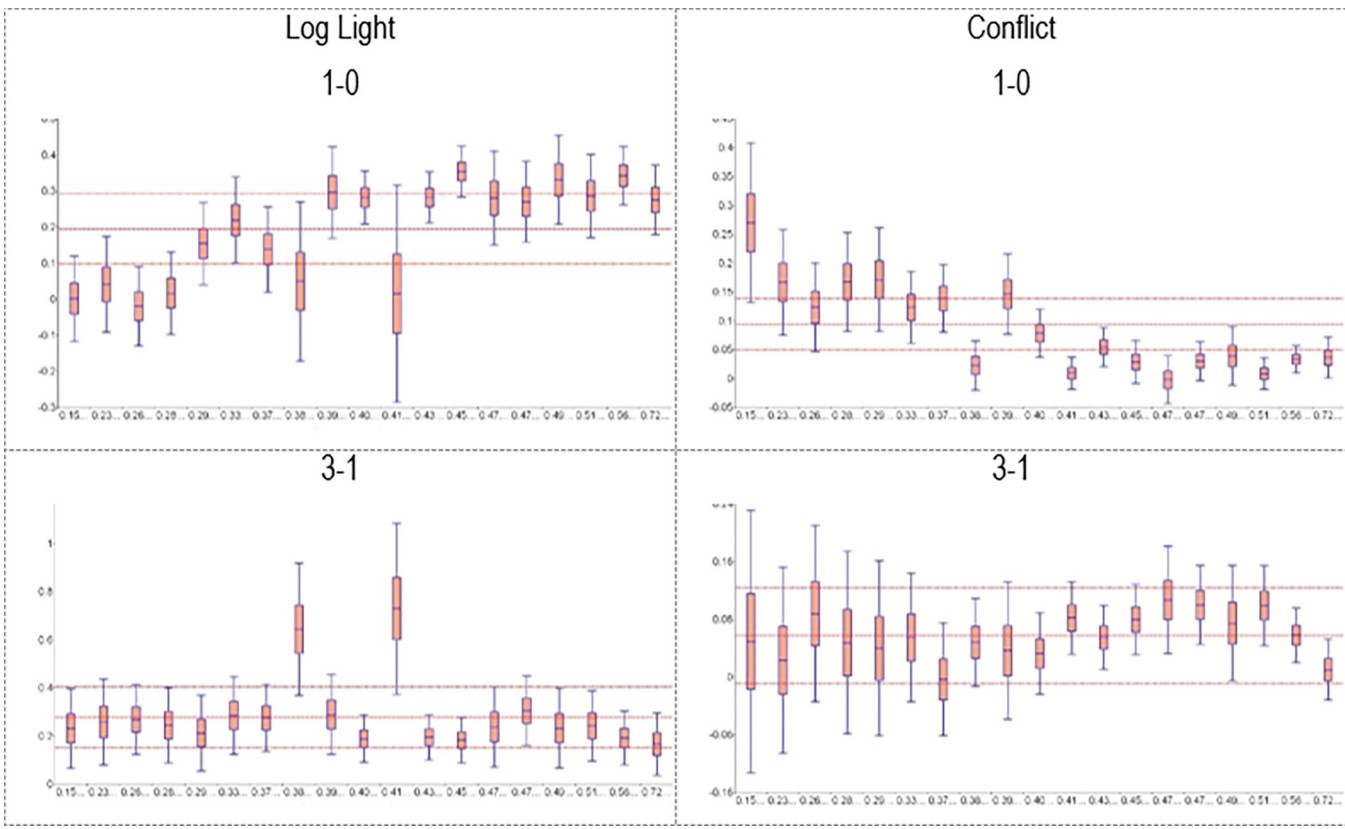

**Fig 3. GATEs for different levels of quality of government (M = 4).** Note: The graphs show the GATEs for the various levels as well as their 90% confidence interval. The horizontal red lines indicate the ATEs and their 90% confidence interval (reported in Panel B of Table 1). Horizontal axes: Values of the variable quality of government. Vertical access: Values of GATEs and ATE.

on economic development and conflict do not vary systematically in institutional quality. This pattern is at odds with the general argument often made based on the work by [2,3].

To better understand this last result, let us turn back to the theoretical model by [2]. Their model suggests that institutions matter for the developmental effects of natural resources because higher resource rents make it more attractive for individuals to specialize in productive activities if institutional quality is high, but in rent seeking activities and fighting if institutional quality is low. We confirm these predicted effects of natural resources on economic development and conflict when focusing on (extensive-margin) differences in mining activities, but not when focusing on (intensive-margin) differences in mineral prices. A possible reason is that changes in mineral prices are typically harder to foresee and less permanent than changes in the presence of industrial mines. Price changes are thus less likely to trigger a widespread reassessment of the individuals' decisions to specialize in either productive activities or in rent seeking and fighting. As institutions play a key role in these specialization decisions (as suggested by [2]), it may not be surprising that institutional quality plays no major role in shaping the developmental effects of higher mineral prices, but a crucial role for the developmental effects of new mining activities.

## Conclusions

Following [2,3], it has become a widely held belief that natural resource rents are a blessing if institutions are strong, but a curse if institutions are weak. In this paper, we have reassessed

the effects of mining and mineral prices on economic development and conflict and, in particular, how these effects depend on institutional quality. We have used data from 3,800 Sub-Saharan African districts and applied [23]'s MCF estimator, which is well suited to analyze causal heterogeneity at various aggregation levels.

We have three main findings as follows. First, mining activities and higher world market prices of locally mined minerals both increase economic development as well as the likelihood of conflict *on average*. Second, the presence of mining activities has more positive effects on local economic development and weaker effects on conflict in countries with higher institutional quality. Third, the effects of higher world market prices of locally mined minerals (rather than mining activities per se) on economic development and conflict vary little in institutional quality but tend to be non-linear: they are typically small for changes from low to intermediate prices, but large for changes from intermediate to high prices.

Our study has important implications for economic policy and the future of research in development economics and related fields. Let us begin with the latter: our study shows that the application of causal machine-learning estimators can lead to novel and more nuanced insights than linear fixed-effects specification. This is also thanks to the possibility to estimate GATEs and to detect non-linearities.

Let us turn to policy implications. Our findings confirm the general argument and original findings of [2,3] that the mining activities have more positive (less negative) effects, the higher the institutional quality. However, our findings also show that this argument does not extend itself to changes in international commodity prices. Therefore, well-meaning local policy makers and international (donor) organizations should indeed prioritize institutional safeguards for countries that start extracting natural resources, but that such safeguards may be less important in case of increasing commodity prices. Future research should carefully disentangle various indicators of institutional quality in order to improve our understanding about which aspects of sound economic and political institutions are most important in ensuring that newly discovered resources are a blessing rather than a curse.

## Supporting information

**S1 Appendix.**
(DOCX)

## Author Contributions

**Conceptualization:** Roland Hodler, Michael Lechner, Paul A. Raschky.

**Data curation:** Paul A. Raschky.

**Formal analysis:** Michael Lechner.

**Methodology:** Michael Lechner.

**Visualization:** Michael Lechner.

**Writing – original draft:** Roland Hodler, Michael Lechner, Paul A. Raschky.

**Writing – review & editing:** Roland Hodler, Michael Lechner, Paul A. Raschky.

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
