## [Decision Letter · Decision Letter 0]

13 Jan 2023

PONE-D-22-33969Institutions and the Resource Curse: New Insights from Causal Machine LearningPLOS ONE

Dear Dr. Raschky,

Thank you for submitting your manuscript to PLOS ONE. After careful consideration, we feel that it has merit but does not fully meet PLOS ONE’s publication criteria as it currently stands. Therefore, we invite you to submit a revised version of the manuscript that addresses the points raised during the review process.

We have acquired three reviews. All three reviewers value the contribution of your paper, but all of them also have suggestions for how to improve the paper and fix weaknesses before it can be considered for publication. Based on my own reading of the paper, the reviewers' comments are constructive and point at multiple areas of potential improvement that should be taken seriously. If you agree to revise your paper, the resubmitted version will be sent back to the same reviewers.

We look forward to receiving your revised manuscript.

Kind regards,

Jerg Gutmann

Academic Editor

PLOS ONE

Reviewers' comments:

Reviewer's Responses to Questions

**Comments to the Author**

1. Is the manuscript technically sound, and do the data support the conclusions?

Reviewer #1: Yes

Reviewer #2: Yes

Reviewer #3: Yes

2. Has the statistical analysis been performed appropriately and rigorously? 

Reviewer #1: Yes

Reviewer #2: Yes

Reviewer #3: Yes

3. Have the authors made all data underlying the findings in their manuscript fully available?

Reviewer #1: Yes

Reviewer #2: Yes

Reviewer #3: Yes

4. Is the manuscript presented in an intelligible fashion and written in standard English?

Reviewer #1: Yes

Reviewer #2: Yes

Reviewer #3: Yes

5. Review Comments to the Author

Reviewer #1: The paper addresses an important research question that has been examined by a battery of studies during the past years and decades. The main innovation of the paper is to employ causal forest estimators to study whether the resource-development nexus depends on the strength of institutions. For this particular setting, I agree with the authors that the causal forest estimator provides some valuable statistical properties. One of the authors is actively involved in developing this approach further, and so quite unsurprisingly, the authors demonstrate excellent command of this technique and use the approach very convincingly. I have no doubt that the results are credible, but the findings are also very much in line with the results of previous studies on the topic. Hence, the contribution to the literature, in this respect, is limited. It was also a bit unclear to me in how far the paper aims to provide a technical contribution (in which case the description of the main method compared to other approaches should be strengthened) or mainly a content-specific contribution (in which case the results should be discussed in much greater detail against the backdrop of previous findings). In general, I think that the methodology employed in the paper can be very valuable also in other settings, and I think that the paper does find results (particularly non-linearities) that we would not typically expect when using traditional methods. These differences should however be discussed in a more systematic way. I also have some additional comments on the paper and the empirical method that I discuss in detail in the attached report.

Reviewer #2: I am glad for the opportunity to read and review the manuscript "Institutions and the Resource Curse: New Insights from Causal Machine Learning". In general, the article attempts to bring a contribution to the literature on the nature of the resource curse from an institutional perspective, by using causal machine learning tools.

In my opinion, the article is very interesting and valuable given the topic and methodology. It may be suitable for publication in PLOS ONE upon a minor revision. I list my concerns regarding the manuscript below (in the order in which they appear in the manuscript):

1. The part of the Introduction on the character of causal machine learning (p. 3) can be restructured as it contains several repetitions in the current form

2. The non-parametric/non-linear estimation strategy is well-justified; the description of the modified causal forest estimator is transparent as well as the relevance of ATEs, GATEs and IATEs

3. The authors can be straightforward about the advantages of using light intensity as a proxy for economic development – especially the availability of data at the subnational level; on the other hand, potential caveats of this approach should be described as well

4. It can be stated that local conflicts may be of various types, intensity and duration, so the disadvantages of using a binary variable in this context should be provided

5. More arguments for choosing the particular geographical and time scope are suggested to be provided

6. The main text should include at least a brief review of the confounding variables, as well as descriptive stats analysis (also to understand better how these factors are included in the models)

7. Given that the authors aim to address the role of institutions in the context of the resource curse, they may consider supplementing the article with XAI tools (e.g. Shapley values or feature importance index) to see the contribution of institutions (constraints on the executive/quality of government) to the model

8. It would be also beneficial to relate the obtained results to the existing literature in a broader manner

9. Proofreading of the manuscript is recommended as it contains some typos, punctuation mistakes and minor stylistic errors

I hope these comments can be useful in improving the paper.

Reviewer #3: Paper: Institutions and the Resource Curse: New Insights from Causal Machine Learning

Journal: PLOS ONE

Manuscript-ID: PONE-D-22-33969

Short summary:

Using district-level panel data from Africa and applying a causal Machine Learning approach, the paper provides novel insights on the effects of natural resouces on economic development and conflict. Thereby, special attention is paid on the question whether these effects depend on the level of institutional quality. The main results are that well-functioning institutions increase the positive effects of mining activities on development and reduces their negative effect on conflict. By contrast, institutional quality plays no role for the effects of mineral prices on these outcomes.

General assessment.

From my perspective, the paper is a valuable contribution to the literature. I also believe that the papers fits to the journal. I have two major comments and some smaller points. You can find them below.

Major comments.

(1) My first set of major comments are related to your measures of institutional quality.

(i) You use the Quality of Government indicator from the ICRG rating by the PRS Group to measure economic institutional quality. I think that this measure is a bad choice for several reasons. First, the way of how the PRS Group produces their measures is a black box as their are no coding rules available. Furthermore, the measure is based on various arbitrary decisions: for instance, there is no theoretical explanation for why "corruption" as well as "law and order" are measured with 0-6 scale, while a 0-4 scale is used to measure bureaucratic efficiency. Another problem is the aggregation procedure that is used to combine the three components. More specificaay, a recent study in the European Economic Review suggests that indices that are based on additive aggregation methods are likely to produce upward-biased estimates (see Gründler and Krieger, 2022).

From my perspective, the popularity of the measures developed by the PRS Group decreased quite a lot over the last years. I even believe that using this data can be regarded as outdated. My suggestions is thus to use another data source. The V-DEM is probably the best choice for your purpose.

(ii) You use a sub-component of the POLITY index to measure political institutional quality. I know that POLITY is widely used, but most people who extensively thought about the measurement of political institution heavily criticize this measure (see e.g.\\ Munck and Verkuilen (2002, Comparative Political Studies), Cheibub et al. (2010, Public Choice), Boese (2019, International Area Studies Review), Gründler and Krieger (2021, European Journal of Political Economy)). Put differently, I think that you should dig deeper into the literature on the measurement of democracy in order to find better alternative.

(iii) From a more conceptual perspective, I wonder why you consider the quality of economic and political institutions separately from each other in your analysis. There is large number of studies, suggesting that economic institutional quality is influenced by the type of political regime.

(iv) Both in the introduction and the conclusion, you stress the policy relevance of your study. In principle, I agree on this point, but I think the paper improves if you disentangle the role of the different aspects of institutional quality. For instance, from a policy maker perspective, it would be nice to know whether fighting off corruption is worse or better than improving the efficiency of the bureaucracy or the independece of the courts. The V-DEM dataset might be helpful here as well.

(2) In your empirical analysis, you try to control for a large number of potential confounders. I wonder a bit about the choice. More specifically, you add a large number of additional measures of institutional quality. I am quite concerned about this approach as some of them even measure the same the variables that you use to study effect heterogeneity. You also add the Polity index as control variable, which is (by construction) depending on the measure of executive constraints. For me, it is not clear why you do this and I wonder whether these additional measures of institutional quality are bad controls and thus bias your main estimates.

Other comments.

(3) You do not find that insitutional quality influences the effects on mineral prices on economic development and conflict. I would like to see some words on why this might be the case, especially as your result is at odds with the papaers by Mehlum et al.

(4) I wonder whether you shoud exclude districts from the control group, as there are some studies that find spillover effects.

(5) On page 3, you write: "We argue that our non-parametric approach is preferable or, at least, a worthwhile alternative, because linear fixed-effects specifications are based on a set of assumptions that may not hold." Of course, this is true, but your approach is also based on assumptions that might be violated. I think you rather write that your approach relaxes some of the assumptions of the linear fixed-effect approach.

(6) For my taste, Section 2 is a bit too long.

(7) In footnote 12, you write that you add 0.01 to the measure of nightlight intensity before taking the log. Given the recent working paper by Chen & Roth ("Log-like? Identified ATEs defined with zero-valued outcomes are

(arbitrarily) scale-dependent"), I wonder whether this approach creates biases.

6. PLOS authors have the option to publish the peer review history of their article (what does this mean?). If published, this will include your full peer review and any attached files.

Reviewer #1: No

Reviewer #2: No

Reviewer #3: No

---

## [Decision Letter · Decision Letter 1]

13 Apr 2023

Institutions and the Resource Curse: New Insights from Causal Machine Learning

PONE-D-22-33969R1

Dear Dr. Raschky,

We’re pleased to inform you that your manuscript has been judged scientifically suitable for publication and will be formally accepted for publication once it meets all outstanding technical requirements.

Kind regards,

Jerg Gutmann

Academic Editor

PLOS ONE

Additional Editor Comments (optional):

Reviewers' comments:

Reviewer's Responses to Questions

**Comments to the Author**

1. If the authors have adequately addressed your comments raised in a previous round of review and you feel that this manuscript is now acceptable for publication, you may indicate that here to bypass the “Comments to the Author” section, enter your conflict of interest statement in the “Confidential to Editor” section, and submit your "Accept" recommendation.

Reviewer #1: All comments have been addressed

Reviewer #2: All comments have been addressed

Reviewer #3: All comments have been addressed

2. Is the manuscript technically sound, and do the data support the conclusions?

Reviewer #1: Yes

Reviewer #2: Yes

Reviewer #3: Yes

3. Has the statistical analysis been performed appropriately and rigorously? 

Reviewer #1: Yes

Reviewer #2: Yes

Reviewer #3: Yes

4. Have the authors made all data underlying the findings in their manuscript fully available?

Reviewer #1: Yes

Reviewer #2: Yes

Reviewer #3: Yes

5. Is the manuscript presented in an intelligible fashion and written in standard English?

Reviewer #1: Yes

Reviewer #2: Yes

Reviewer #3: Yes

6. Review Comments to the Author

Reviewer #1: (No Response)

Reviewer #2: (No Response)

Reviewer #3: The authors replied to all my requests in a satisfactory manner. Put differntly, I like the revised version of the paper and think that it is ready for publication.

7. PLOS authors have the option to publish the peer review history of their article (what does this mean?). If published, this will include your full peer review and any attached files.

Reviewer #1: No

Reviewer #2: No

Reviewer #3: No

---

## [Editor Report · Acceptance letter]

16 May 2023

PONE-D-22-33969R1 

Institutions and the Resource Curse: New Insights from Causal Machine Learning 

Dear Dr. Raschky:

I'm pleased to inform you that your manuscript has been deemed suitable for publication in PLOS ONE. Congratulations! Your manuscript is now with our production department. 

Kind regards, 

on behalf of

Prof. Dr. Jerg Gutmann 

Academic Editor

PLOS ONE